# Association between Biomarkers (VEGF-R2, VEGF-R3, VCAM-1) and Treatment Duration in Patients with Neuroendocrine Tumors Receiving Therapy with First-Generation Somatostatin Analogues

**DOI:** 10.3390/biomedicines11030842

**Published:** 2023-03-10

**Authors:** Violetta Rosiek, Ksenia Janas, Beata Kos-Kudła

**Affiliations:** Department of Endocrinology and Neuroendocrine Tumors, Department of Pathophysiology and Endocrinology, Medical University of Silesia, 40-514 Katowice, Poland

**Keywords:** somatostatin analogues, neuroendocrine tumors, VEGF-R2, VEGF-R3, VCAM-1

## Abstract

Angiogenic factors (AF) promote vascular formation and may thus support neuroendocrine tumour (NET) development. This study aimed to assess AF serum level changes in NET patients treated with prolonged-acting somatostatin analogues (SSAs). The study enrolled 49 healthy volunteers (Group A) and 56 NET patients: treatment naïve (Group B) and after-SSA treatment in various periods (months): under 12 (Group C), 13–24 (Group D), 25–36 (Group E), 37–60 (Group F), and over 60 months (Group G). The serum vascular endothelial growth factor receptors 2, 3 (VEGF-R2, VEGF-R3), and vascular cell adhesion molecule-1 (VCAM-1) concentrations were tested using the ELISA. We noted significant differences in the concentrations of VEGF-R2, VEGF-R3, and VCAM-1 depending on the SSA treatment duration (*p* < 0.001). In the studied AFs, the highest decreasing levels of VEGF-R2 were observed after two years of therapy. However, monitoring VEGF-R2, VEGF-R3, and VCAM-1 during SSA treatment did not allow for the identification of good responders for this kind of therapy. Therefore, these biomarker measurements were not helpful in assessing SSA treatment effectiveness in NET patients.

## 1. Introduction

The majority of neuroendocrine tumours/neoplasms (NETs/NENs) are well-differentiated neoplasms that present overexpression of somatostatin receptors. For this reason, first-line therapy is systemic treatment with prolonged-acting somatostatin analogs (SSAs). SSA is the treatment for patients with functional (antisecretory effect) and non-functional NETs (NF-NETs) (antiproliferative effect as well as an anti-angiogenic action) [1,2,3,4]. SSAs reduce symptoms such as diarrhea; flush, by decreasing the secretion of biologically active substances and hormones and are used in NET patients with stable (SD) or progressive disease (PD) and a low proliferation index (Ki-67 ≤ 10%). According to the Consensus Guidelines for the Standards of Care in Neuroendocrine Neoplasms of the Polish Network of Neuroendocrine Tumours (PNNT) experts (2022) [1], as well as the European Neuroendocrine Tumour Society (ENETS) (2017) [5] and European Society for Medical Oncology (ESMO) (2020) [6], SSA therapy in studied NET patients was administered every four weeks (lanreotide 120 mg s.c. or octreotide 30 mg i.m.). 

One of the important mechanisms of cancer development is angiogenesis, which is essential for cancer progression, while lymphangiogenesis has also been shown to be involved in tumour metastasis. NETs are hypervascular tumours that have an expression of vascular endothelial growth factors (VEGFs) as strong tumour angiogenesis regulators [7] and are essential factors for metastatic spread [8]. VEGF is a central player in regulating tumour angiogenesis, which is regulated via three receptor tyrosine kinases (VEGFR-1–3) [9,10].

The VEGF/VEGF receptor pathway plays an important role in blood and lymphatic vessel development and the pathogenesis of cancer disease [11,12,13,14,15]. VEGF receptor 2 (VEGF-R2) signalling axes play a pivotal role in angiogenesis [16], while VEGF receptor 3 (VEGF-R3) plays an important role in lymphangiogenesis [17]. Cigrovski-Berković M. et al. examined the VEGF gene/polymorphisms in gastro-entero-pancreatic NET tumorigenesis. The results suggest that the highest serum VEGF levels correlate with lymph node metastases [10]. However, higher microvascular density is a bad prognostic factor in most carcinomas. It seems to be a good marker for pancreatic NETs (pNETs). In contrast to other kinds of neoplasms, pNETs are highly vascularised but poor angiogenic tumours, and if they progress, VEGF expression is lost and microvascular density is significantly lower [18]. Vascular cell adhesion molecule-1 (VCAM-1) is a transmembrane molecule that mediates the adhesion of immune cells to the vascular endothelium during inflammation and is related to the development of malignant neoplasms, such as breast cancer, melanoma, and renal clear cell carcinoma. VCAM-1 can also promote tumour cell invasion and metastasis [19].

Molecular examinations to stratify neoplasm patients for targeted treatment are needed, but their high cost and technical barriers limit the usage of these tests. Other investigators have applied deep learning-based studies to predict molecular test results from digitised images of tissue slides [20]. Therefore, in this study, we assessed serum level changes of angiogenic factors (AFs) (VEGF-R2, VEGF-R3, and VCAM-1) in NET patients to determine whether these factors are helpful in determining the effectiveness of treatment with SSAs (lanreotide, octreotide). A significant association between AFs changes and disease response could facilitate a decrease in the frequency of monitoring CT or MRI scans for therapy follow-up and a reduction in radiation exposure for NET patients.

We aimed to check if these AFs measurements were warranted, both in the decision to begin SSA treatment and in the follow-up to the response to this therapy, and in selecting the NEN patient subgroup in whom this therapy gives the most benefits. 

## 2. Materials and Methods

### 2.1. Cohorts 

The study comprised 49 healthy subjects (Group A) and 56 NET treatment naïve patients (Group B) after SSA treatment for various periods (months): under 12 (Group C), 13–24 (Group D), 25–36 (Group E), 37–60 (Group F), and over 60 months (Group G). At the time of initiation of SSA therapy, 56 patients suffered from a NET. After the subsequent months of SSA therapy, some patients developed disease progression and took the next treatment step (second- or third-line treatment). Therefore, the number of patients in the groups decreased (Group C, n = 55; Group D, n = 46; Group E, n = 35; Group F, n = 26; and Group G, n = 22).

We included adults over the age of 18 years with a histologically confirmed diagnosis of NET and healthy controls, after obtaining written informed consent from each study participant. 

Medical record data and VEGF-R2, VEGF-R3, and VCAM-1 serum concentration measurements of SSA-treated NET patients were used for analysis. SSAs were administered to NET patients every four weeks: 42/56 patients received Somatuline Autogel 120 mg s.c. (75%), and 14/56 patients received Sandostatin LAR 30 mg i.m. (25%). No one was treated with peptide receptor radionuclide therapy (PRRT). 

The control group comprised members of hospital personnel and non-affected patients attending the occupational medicine clinic. All controls were asymptomatic, in good health, and known to have an absence of malignancy.

Exclusion criteria for both groups were other malignancies and heart, renal, or liver failure.

All subjects gave their informed consent for inclusion before they participated in the study. The study was conducted in accordance with the Declaration of Helsinki, and the protocol was approved by the Ethics Committee of the Medical University of Silesia, Poland (no. KNW/0022/KB1/130/I/15 and PCN/0022/KB1/97/I/19/20).

The study examinations were provided at the Department of Endocrinology and Neuroendocrine Tumours, the ENETS Centre of Excellence (CoE), and the Endocrinology Specialist Outpatient Clinic in Katowice. 

### 2.2. Diagnostic and Analytical Methods

Venous blood samples were taken from all NET patients and the control group (healthy subjects) and were centrifuged at 3000 rpm for 10 min. Then, the serum was stored at a temperature of −80 °C for further analysis. The serum samples for VEGF-R2, VEGF-R3, and VCAM-1 measurements were collected before and after SSA treatment. The levels of VEGF-R2, VEGF-R3, and VCAM-1 in the blood serum of all study participants were assessed using the ELISA method. Serum VEGF-R2 and VCAM-1 were determined using Quantikine ELISA, provided by R&D Systems (Minneapolis, MN, USA), and serum VEGF-R3 using Platinum ELISA, provided by Affymetrix eBioscience, according to the manufacturer’s protocol. The results of the VEGF-R2 and VCAM-1 concentrations were presented in ng/mL, whereas the level of VEGF-R3 was in pg/mL.

Metrics of the studiedAFs:

VEGF-R2 metrics: The sensitivity of the method was 4.6 pg/mL; expected values (range) were 6420–14,501 pg/mL; intra-assay and inter-assay precisions were 2.9–4.2% and 5.7–7.0%, respectively.

VEGF-R3 metrics: The sensitivity of the method was 0.03 ng/mL; expected values (range) were 33–167 ng/mL; intra-assay and inter-assay precisions were 2.4–14.1% and 2.6–7.2%, respectively.

VCAM-1 metrics: The sensitivity of the method was 0.6 ng/mL; expected values (range) were 349–991 ng/mL; intra-assay and inter-assay precisions were 2.3–3.6% and 5.5–7.8%, respectively. 

### 2.3. Radiological Evaluation of NET Disease

To monitor NET disease and evaluate the presence of distant metastases or detect progression, most NET patients had undergone functional imaging (by [^68^Ga]Ga-DOTATATE positron emission tomography (PET)/computed tomography (CT)) and anatomical scans. Standard anatomical imaging with CT or magnetic resonance imaging (MRI) was performed every 6 months during the follow-up, and [^68^Ga]Ga PET/CT was usually performed every year. All images were evaluated by experienced specialists in radiology and/or nuclear medicine.

### 2.4. Histological Diagnosis

All NET patients had histologically confirmed NET diagnoses. For diagnostics, we used the surgical specimens, polyps with NET tissue or biopsy (the highest sensitivity was the core needle biopsy [21]), and all specimens were evaluated by H&E staining and immunohistochemistry. The tumour grade was assessed in accordance with the TNM 8th edition classification of gastrointestinal neuroendocrine neoplasms (GEP-NEN), according to the recommendations of the American Joint Committee on Cancer/Union for International Cancer Control and the World Health Organization (WHO) classification from 2017. A group of highly differentiating neoplasms was distinguished. The following categories were created based on the Ki-67 proliferation index: NET G1 (with the Ki-67 less than 3%, there were 38/56 subjects, Ki-67 1% in 19/56 subjects, and Ki-67 2% in 19/56 subjects), and NET G2 (with the Ki-67 proliferation index ranging from 3% to 20%, there were 18/56 subjects, Ki-67 3% in 8/56; Ki-67 4% in 2/56; Ki-67 5% in 5/56; Ki-67 8% in 1/56; and Ki-67 10% in 2/56 subject,). The project did not include the NET G3 and NEC (group of poorly differentiated cancers).

According to the 2015 WHO classification of lung and pleural neoplasms from NENs of the respiratory system (BP-NEN-broncho-pulmonary-NEN), we created two groups: typical carcinoids (TCs) (1/4 subjects) and atypical carcinoids (ACs) (3/4 subjects). Our study did not include large-cell neuroendocrine carcinoma (LCNEC) or small-cell lung cancer (SCLC).

### 2.5. Statistical Analyses

Statistical analyses were carried out using STATISTICA version 13.36.0 (StatSoft) software. The distribution of the data was determined by the Kolmogorov–Smirnov test. Data are presented as median and interquartile ranges for nonparametric data. The comparison of VEGF-R2, VEGF-R3, and VCAM-1 concentrations between the NET patients and control groups was performed using the Mann–Whitney U-test. Furthermore, an intergroup analysis (SSA-treated NET patients) was undertaken using a nonparametric test (Friedman test) for multiple samples. The Wilcoxon test for paired samples was used where appropriate. To investigate the prognostic value of VEGF-R2, VEGF-R3, and VCAM-1 in predicting SSA-treatment response in NET patients, receiver operating characteristic (ROC) curves were plotted, and the sensitivity, specificity, and area under the curve (AUC) were calculated. For correlation analysis, *p* values and correlation coefficients (r) were calculated using Spearman’s correlation test. Results were considered significant at *p* < 0.05.

## 3. Results

The demographic, biochemical, and clinical characteristics of the participants recruited for the study (NET patients and controls) are presented in Table 1. The NET patient cohort consisted of 54% males and 46% females, with a median age of 64. All patients were diagnosed with well-differentiated NETs; thirty-eight patients had G1 NETs, while eighteen patients had G2 NETs. The majority of them had advanced disease stages (61%) III and IV of TNM (43% of NET patients had distant metastases) at the time of diagnosis. The most common primary site location was the pancreas (42%). Twenty percent of these patients had F-NETs. 

The control subjects comprised, in contrast to NET patients, only 29% males and 71% females, with a median age of 54 years (10 years less than NET patients).

The VEGF-R3 and VCAM-1 concentrations were significantly elevated in the NET cohort compared to controls. There were no significant differences in VEGF-R2 levels in both groups (Table 2).

### 3.1. Angiogenic Factors (VEGF-R2, VEGF-R3, and VCAM-1) in SSA-Treated NET Patients

The comparison of VEGF-R2, VEGF-R3, and VCAM-1 in NET patients before and after SSA treatment is shown in Appendix A. 

The Friedman test (Appendix A) showed significant differences in VEGF-R2, VEGF-R3, and VCAM-1 level intergroup analysis (between SSA-treated NET patients and treatment naïve NET patients).

In the next step, we used the Wilcoxon signed rank test to test two dependent samples (before and after SSA therapy), and thus we analysed whether there was a significant difference between the levels of these angiogenic factors (VEGF-R2, VEGF-R3, and VCAM-1). Additionally, the Wilcoxon test showed that these differences were statistically significant (*p* < 0.05) (Table 3). 

In the NET patient group, we had statistically significant evidence that the median differences in VEGF-R2 and VCAM-1 levels were significant. There were significant differences in VEGF-R2 (Figure 1a) and VCAM-1 levels in all groups after SSA treatment compared to before SSA therapy (Table 3, Figure 1a,c). Given the concentration of VEGF-R3, in the group of patients treated for 12 and 24 months, VEGF-R3 concentrations were similar to those before treatment and slightly increased. It was only after 2 years of SSA therapy that they began to decrease and were significantly lower than those before treatment (Groups E, F, and G) (Table 3, Figure 1b). 

Moreover, regarding Spearman’s correlation assessment of the relationship between VEGF-R2, VEGF-R3, VCAM-1, and the duration of treatment (Appendix A), a negative, statistically significant correlation was shown between VEGF-R2 and the duration of treatment. With the increase in the treatment period, a decrease in VEGF-R2 levels was observed (Figure 2). Regarding Spearman’s correlation assessment of the relation of VEGF-R3 and VCAM-1 to the duration of treatment, it was also not statistically significant (Appendix A).

In the third step, the ROC analysis and AUC were used to assess the capacity of AF to predict SSA-treatment response based on AF level changes. AUC analyses could only differentiate SSA-non-treated (Group B) from SSA-treated NET patients (Groups E and G) for VEGF-R2. Although significant (*p* < 0.05), it should be noted that all AUCs below 0.55 and less would be considered poor biomarkers (Appendix A).

### 3.2. Assessment of Angiogenic Factors (VEGF-R2, VEGF-R3, and VCAM-1) Levels According to Tumour Grade

In patients with tumor grade G1, the median VEGF-R2, VEGF-R3, and VCAM-1 values were not significantly different from those in the group with tumor grade G2 (*p* > 0.05). A similar tendency was observed in the analysis of the final levels of these biomarkers, namely in the G1 group, where the median final VEGF-R2, VEGF-R3, and VCAM-1 serum concentrations were similar to those in the tumor grade G2 (*p* > 0.05) (Appendix A).

### 3.3. Assessment of Angiogenic Factors (VEGF-R2, VEGF-R3, and VCAM-1) Levels According to the Disease Stage

In the group of patients with disease progression, the median angiogenic factors (VEGF-R2, VEGF-R3, and VCAM-1) levels were similar compared with the group with disease stabilization (*p* > 0.05). In the analysis of final angiogenic factors (VEGF-R2, VEGF-R3, and VCAM-1) concentration in the group with disease stabilization, the median was also similar compared with the group with disease progression (*p* > 0.05). The results of the angiogenic factors (VEGF-R2, VEGF-R3, and VCAM-1) concentration assessment are presented in Appendix A.

## 4. Discussion

Angiogenesis is one of the core hallmarks of all cancers [22,23]. NETs are highly angiogenic and consequently densely vascularized [24]. Recently, a number of markers have been studied to find the correlation with NET status. According to the latest guidelines (ENETS 2019, ESMO 2020, PNNT 2022), somatostatin analogues are a well-established first-line therapy in functional and non-functional NETs. Therefore, our study tried to assess angiogenic factor serum level changes in NET patients treated with prolonged-acting somatostatin analogues (SSAs). 

CgA is most commonly used for therapy monitoring of patients with well-differentiated NET patients treated with SSAs. The prospective, multicenter study of GEP-NET patients by Dam G. et al. [25] showed a weak association between a change in plasma CgA and a change in tumor burden.

Recently, Puliani G. et al. demonstrated the relation between angiogenic markers and gastro-entero-pancreatic (GEP) and pulmonary NET morphology and staging. The study showed higher concentrations of angiopoietin 1 and 2 (ANG1, ANG2), soluble TIE2 (sTIE2), and prokinectin 2 (PROK2) in NET patients compared to healthy controls. Furthermore, the level was higher in poorly differentiated NENs and more advanced (stages 3–4) diseases. The authors suggested a potential role for ANG2 and PROK2 as markers of progression in pulmonary and GEP-NENs, as the concentration of those markers was significantly higher in PD compared to SD [26]. Another study by Sesti F. et al. showed higher levels of ANG1 and ANG2 and a higher count of Tie2-expressing monocytes in GEP-NEN patients compared to controls, which suggests a connection between immunity and angiogenic pathways in NENs [27].

Our previous paper [28] aimed to evaluate the serum VEGF and vascular endothelial growth factor receptor 1 (VEGF-R1) concentration changes in patients with NET who were also treated with first-generation long-acting SSAs. These AFs (VEGF and VEGF-R1) have limited use in the assessment of SSA treatment effectiveness in NET.

VEGF signalling pathways in NENs are one of the most studied angiogenic factors. Controversial conclusions have been obtained from the analysed data [29]. Some studies showed higher VEGF levels in NET patients compared to healthy controls, however, with a lack of correlation between grading and aggressiveness [10,30,31]. Pavel et al. analysed the connection between tumour growth and the release of angiogenic factors, and they showed a decrease in VEGF concentration upon octreotide therapy introduction and an increase in cases of progression [30]. On the other hand, Zurita et al. revealed that baseline VEGFR-2 levels are predictive of better survival in pancreatic well-differentiated NETs [32].

A few other studies have shown no statistical significance of VEGF concerning the study objectives [32,33,34]. 

In the current analysis, given the VEGF-R2 and VEGF-R3 level changes during SSA treatment, we confirmed significant differences between NET patients before and after SSA treatment for VEGF-R2 (the concentration of VEGF-R2 decreases), VCAM-1 (the concentration of VEGF-R2 decreases), and VEGF-R3 (a statistically significant decrease was observed in patients after greater than 2 years of treatment) compared to Group B (NET patients before treatment). 

However, we did not find a correlation between the radiologic tumour responses (SD, PD) and the change in serum angiogenic factors of NEN patients treated with SSA.

An increasing amount of data shows that VCAM-1 is closely associated with tumour angiogenesis and metastasis. VCAM-1 is abnormally expressed in gastric cancer, renal clear cell carcinoma, melanoma, breast cancer, glioma, and other malignant tumours. It is also unfavourably correlated with the prognosis [35]. Ding Y.B. et al. studied gastric cancer. They revealed that VCAM-1-positive tissue has a higher micro-vessel density than VCAM-1-negative tissue [36]. Other reports demonstrate that serum VCAM-1 concentration correlates with the micro-vessel density of breast cancer and may be a substitutive marker of angiogenesis [37]. Overexpression of VCAM-1 in colorectal cancer cells is closely associated with the invasive and aggressive clinical characteristics and poor prognosis of colorectal cancer patients [19]. A recent study analysing the association between treatment and systemic inflammation in acromegaly showed interesting data. VCAM-1 concentration was highest in untreated patients, and VCAM-1 concentration was reduced in treated subjects, i.e., with somatostatin analogues [38]. A review by Kong et al. highlighted the emerging potential of VCAM-1 as a novel therapeutic target in immunological disorders and cancer [35]. Our study noted statistically significant differences between the serum VCAM-1 level in treated and non-treated NET patients, but further analysis revealed no clinical usefulness for assessing SSA treatment effectiveness.

Tyrosine kinase inhibitors (TKI), which are widely used in the management of NETs [1,5,6] seem to have a similar effect on VEGF-R concentration. A recent study showed the ability of a particular TKI to reduce VEGF/VEGF-R2 levels [39]. Further studies are required to establish whether this effect is multiplied by combined SSA + TKI therapy in neuroendocrine tumours.

## 5. Conclusions

In this prospective one-center study of NET patients treated with SSA, monitoring angiogenic factors (VEGF-R2, VEGF-R3, and VCAM-1) serum levels did not help identify good responders for this therapy. There was no association between a change in serum angiogenic factors and tumour response.

## 6. Limitations of the Study

The main study limitation is the heterogeneity and variable number of the SSA-treated NET patient subgroup. Furthermore, our analysis was performed on subjects treated with lanreotide or octreotide (in non-equal proportions). The correlation of marker levels with radiologic tumor response is lacking, and the number of samples is limited. So, further studies in larger patient cohorts are warranted.

## Figures and Tables

**Figure 1 biomedicines-11-00842-f001:**
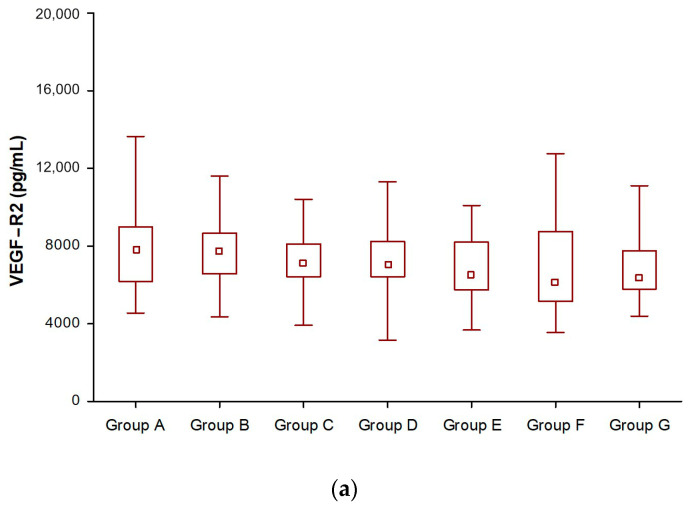
Changes in vascular endothelial growth factor receptor 2 (VEGF-R2) (**a**), vascular endothelial growth factor receptor 3 (VEGF-R3) (**b**), and vascular cell adhesion molecule-1 (VCAM-1) (**c**) levels during somatostatin analogue (SSAs) treatment in patients with neuroendocrine tumors.

**Figure 2 biomedicines-11-00842-f002:**
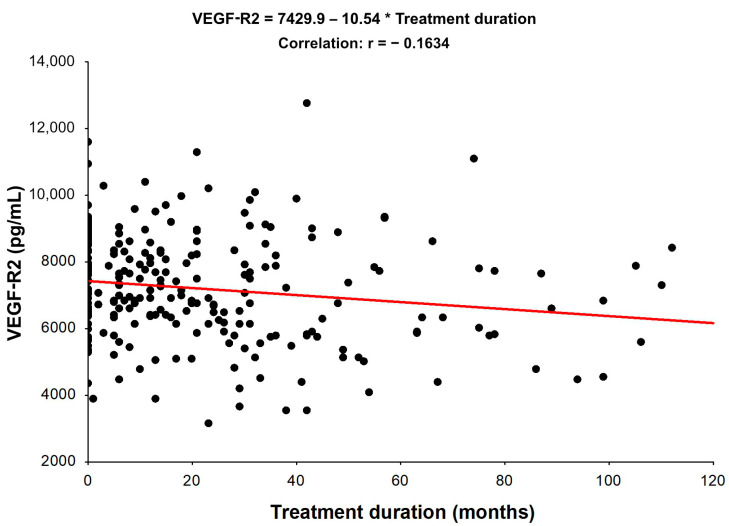
Spearman’s Rank Correlations. Spearman’s coefficients of the relationships among analyzed variables in NET patients: VEGF-R2 vs. Treatment duration (the presented correlation is statistically significant, *p* < 0.05).

**Table 1 biomedicines-11-00842-t001:** Characteristics of the neuroendocrine tumor (NET) patients and the healthy controls.

	NET Patients	Healthy Controls
Number of patients (n)	56	49
Gender		
Male	30 (54%)	14 (29%)
Female	26 (46%)	35 (71%)
Age (years)		
Mean/Median	61.57/64.30	53.36/54.50
Tumour grade		
G1	38 (68%)	N/A
G2	18 (32%)	
Disease stage		
I + II	22 (39%)	N/A
III	10 (18%)	
IV	24 (43%)	
Disease stage		
SD	10 (18%)	N/A
PD	24 (43%)	
Disease extent-metastases		
yes	34 (61%)	N/A
no	22 (39%)	
NET site:		
Gastrointestinal	21 (38%)	
Pancreatic	23 (42%)	N/A
Unknown	6 (11%)	
Lung/Thymus	4/1 (9%)	
Functionality status:		
NF-NET	45 (80%)	N/A
F-NET: CS/Glucagonoma	11: 10/1 (20%)	
Previous surgery		
yes	29 (52%)	N/A
no	27 (48%)	
Kind of SSAs therapy:		
Sandostatin LAR 30 mg	14 (25%)	N/A
Somatuline Autogel 120 mg	42 (75%)	

Data are shown as median, mean, number, and percentage (%). Abbreviations: N/A, not applicable; NF-NET, non-functioning NET; F-NET, functioning NET; CS, carcinoid syndrome; PD, progressive disease; SD, stable disease; SSAs, somatostatin analogues.

**Table 2 biomedicines-11-00842-t002:** Comparison of the studied factors in patients with neuroendocrine tumours (NET) and healthy controls.

Variable	NET Patients	Healthy Controls	*p*-Value *
VEGF-R2 (median/IR)	(7717/6567–8662)	(7793/6186–8980)	NS
VEGF-R3 (median/IR)	(65/56–73)	(41/27–53)	<0.001
VCAM-1 (median/IR)	(667/595–838)	(479/400–565)	<0.001

* Mann–Whitney test; Abbreviations: VEGF-R2, vascular endothelial growth factor receptor 2; VEGF-R3, vascular endothelial growth factor receptor 3; VCAM-1, vascular cell adhesion molecule-1; IR, interquartile range; NS, not significant.

**Table 3 biomedicines-11-00842-t003:** The Wilcoxon matched pairs test of vascular endothelial growth factor receptor 2 (VEGF-R2), vascular endothelial growth factor receptor 3 (VEGF-R3), and vascular cell adhesion molecule-1 (VCAM-1) for SSAs-treated NET patients.

Matched Pairs of Variables	z	*p*
VEGF-R2 Group B & VEGF-R2 Group C	6.45	<0.001
VEGF-R2 Group B & VEGF-R2 Group D	5.73	<0.001
VEGF-R2 Group B & VEGF-R2 Group E	5.16	<0.001
VEGF-R2 Group B & VEGF-R2 Group F	4.28	<0.001
VEGF-R2 Group B & VEGF-R2 Group G	4.11	<0.001
VEGF-R3 Group B & VEGF-R3 Group C	0.23	NS
VEGF-R3 Group B & VEGF-R3 Group D	1.08	NS
VEGF-R3 Group B & VEGF-R3 Group E	5.14	<0.001
VEGF-R3 Group B & VEGF-R3 Group F	4.46	<0.001
VEGF-R3 Group B & VEGF-R3 Group G	3.72	<0.001
VCAM-1 Group B & VCAM-1 Group C	5.21	<0.001
VCAM-1 Group B & VCAM-1 Group D	2.21	0.027
VCAM-1 Group B & VCAM-1 Group E	2.21	0.027
VCAM-1 Group B & VCAM-1 Group F	3.92	<0.001
VCAM-1 Group B & VCAM-1 Group G	4.11	< 0.001

Abbreviations: see Table 2.

## Data Availability

All data are available upon any reasonable request.

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
