# Peer review of "Association between Biomarkers (VEGF-R2, VEGF-R3, VCAM-1) and Treatment Duration in Patients with Neuroendocrine Tumors Receiving Therapy with First-Generation Somatostatin Analogues"

_biomedicines, 2023, doi:10.3390/biomedicines11030842_

Round 1
Reviewer 1 Report
Thank you for this interesting work on monitoring plasmatic angiogenic markers to evaluate treatment response to somatostatin analogues in patients with NETs. Nevertheless, I have some important comments:
Major comments:
1/Somatostatine analogues are indeed a first-line treatment in advanced WDNET for a particular group of patients, it would be relevant to specify these conditions in the introduction section
2/Also, the introduction is missing some litterature for NET tumors regarding the role of angiogenic marker (it is all in the discussion section). Why were these markers in particular chosen for this study?
3/Could you give us more information on NET patients included, such as Ki-67 index? We need the information regarding which ASST was used (proportions) and at what frequency and dose.
4/Why would you also include stage I patients? According to which TNM classification? Why were they not operated on and put on somatostatin analogues? Could there not be a difference on marker levels depending on tumor volume and the existence of distant metastases ?
5/Also, we would like to have the correlation of marker levels with radiologic tumor response and not just treatment duration under analogues:
-indeed, it is unclear to me if some patients might have progressed under analogues but kept them for a functional syndrome?
-also, there can be patients with completely stable diseases and others that progress discretely but stay stable within RECIST criteria
-it would be also interesting to have information on what types of imaging techniques were used to monitor patients: CT? MRI? functional imaging? the sensitivity being quite different to detect progression or distant metastases
6/It is hard to conclude on 56 patients. It would have been interesting to have a bigger group of patients and therefore study these markers in groups of patients with the same primary tumor location, and in particular gastrointestinal vs pancreatic tumors.
7/Where are the results for the control group/group A in the manuscript??
Minor comments :
1/Please have a native English speaker double check the English language for the whole document, there are many sentences that are not correctly constructed which makes it hard for the reader to understand.
For instance, it is hard at first to understand that patients in group C to G are the same patients than in group B.
Also there are some mistakes such as:
*Abstract :
Line 23 : VERF-R2=>VEGR-R2
2/Many statistical tests that are performed in the supplementary material are not mentioned in the statistical analysis, and there is no justification on why performing all these different tests.
Reviewer 2 Report
Authors aimed to assess AF serum level changes in NET patients treated with prolonged-acting somatostatin analogues (SSA).
There are several concerns to be addressed
1. Why were healthy volunteers enrolled?
2. Regarding the duration of SSA, if NETs progress during NET therapy, is it continued? If not, the association between duration and level of AF might have a bias.
3. If eligible, please, show the pathological characteristics.The exploration of correlation between pathology and serum marker/clinical outcome might be interesting.
4. Please, cite the related articles; PMID: 35443570 and PMID: 33317239.
Round 2
Reviewer 1 Report
The authors have addressed most of my previous comments
Please add in the limitations section:
-that the correlation of marker levels with radiologic tumor response is lacking
-the number of the sample is limited and larger studies are needed
Reviewer 3 Report
The authors have addressed most of my previous comments, however the paper lack of correlation with radiological response that limited the novelty and the interest to the readers.
I suggest to improve the paper adding this aspect in this paper.
Moreover, the paper could be improved with histological data regarding the expression of VEGFR2-3 e VCAM1 and their correlation with circulating biomarkers.
Round 3
Reviewer 3 Report
The paper have been improved
